# Neuromechanical Autoencoders: Learning to Couple Elastic and Neural Network Nonlinearity

**Deniz Oktay, Mehran Mirramezani, Eder Medina, Ryan P. Adams**
Department of Computer Science
Princeton University
{doktay,mehranmir,em2368,rpa}@princeton.edu

## Abstract

Intelligent biological systems are characterized by their embodiment in a complex environment and the intimate interplay between their nervous systems and the nonlinear mechanical properties of their bodies. This coordination, in which the dynamics of the motor system co-evolved to reduce the computational burden on the brain, is referred to as "mechanical intelligence" or "morphological computation". In this work, we seek to develop machine learning analogs of this process, in which we jointly learn the morphology of complex nonlinear elastic solids along with a deep neural network to control it. By using a specialized differentiable simulator of elastic mechanics coupled to conventional deep learning architectures—which we refer to as neuromechanical autoencoders—we are able to learn to perform morphological computation via gradient descent. Key to our approach is the use of mechanical metamaterials—cellular solids, in particular—as the morphological substrate. Just as deep neural networks provide flexible and massively-parametric function approximators for perceptual and control tasks, cellular solid metamaterials are promising as a rich and learnable space for approximating a variety of actuation tasks. In this work we take advantage of these complementary computational concepts to co-design materials and neural network controls to achieve nonintuitive mechanical behavior. We demonstrate in simulation how it is possible to achieve translation, rotation, and shape matching, as well as a "digital MNIST" task. We additionally manufacture and evaluate one of the designs to verify its real-world behavior.

## 1 Introduction

Mechanical intelligence, or morphological computation (Paul, 2006; Hauser et al., 2011), is the idea that the physical dynamics of an actuator may interact with a control system to effectively reduce the computational burden of solving the control task. Biological systems perform morphological computation in a variety of ways, from the compliance of digits in primate grasping (Jeannerod, 2009; Heinemann et al., 2015), to the natural frequencies of legged locomotion (Collins et al., 2005; Holmes et al., 2006; Ting & McKay, 2007), to dead fish being able to "swim" in vortices (Beal et al., 2006; Lauder et al., 2007; Eldredge & Pisani, 2008). Both early (Sims, 1994) and modern (Gupta et al., 2021) work have used artificial evolutionary methods to design mechanical intelligence, but it has remained difficult to design systems *de novo* that are comparable to biological systems that have evolved over millions of years. We ask:

*Can we instead learn morphological computation using gradient descent?*

Morphological computation requires that a physical system be capable of performing complex tasks using, e.g., elastic deformation. The mechanical system's nonlinear properties work in tandem with neural information processing so that challenging motor tasks require less computation. To learn an artificial mechanically-intelligent system, we must therefore be able to parameterize a rich space of mechanisms with the capability of implementing nonlinear physical "functions" that connect input forces or displacements to the desired output behaviors. There are various desiderata for such a

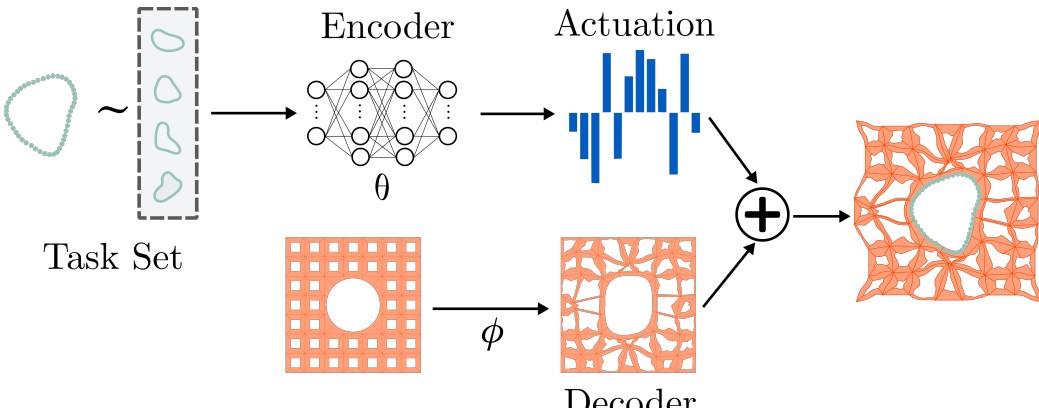

Figure 1: A schematic depiction of a neuromechanical autoencoder. A neural encoder is parameterized by $\theta$, while a mechanical decoder has geometry (morphology) parameterized by $\phi$. A task is sampled from a distribution and is fed into the neural network encoder. The neural network produces actuations which displace the mechanical structure to perform the task, in this case being shape matching (Section 3.2). Given a task loss, $\theta$ and $\phi$ are optimized jointly by gradient descent.

mechanical design space: 1) it must contain a wide variety of structures with complex nonlinear elastic deformation patterns; 2) its parameters should be differentiable and of fixed cardinality; and 3) the designs should be easily realizable with standard manufacturing techniques and materials. These characteristics are achieved by mechanical metamaterials.

Metamaterials are structured materials that have properties unavailable from natural materials. Although metamaterials are often discussed in the context of electromagnetic phenomena, there is substantial interest in the development of *mechanical* metamaterials in which geometric heterogeneity achieves unusual macroscopic behavior such as a negative Poisson's ratio (Bertoldi et al., 2010). In biological systems, morphological computation often takes the form of sophisticated nonlinear compliance and deformation, resulting in a physical system that is more robust and easier to control for a variety of tasks (Paul, 2006; Hauser et al., 2011), This type of behavior is typically not present in off-the-shelf robotic systems and is difficult to design *a priori*. Mechanical metamaterials, on the other hand, offer a platform for mechanically-intelligent systems using relatively accessible manufacturing techniques, such as 3-D printing.

The mechanical metamaterials we explore in this paper are *cellular solids*: porous structures where different patterns of macroscopic pores can lead to different nonlinear deformation behaviors. By constructing a solid with a large number of such pores, and then parameterizing the pore shapes nonuniformly across the solid, it is possible to achieve a large design space of nonlinear mechanical structures while nevertheless having a differentiable representation of fixed cardinality. The key to modern machine learning has been the development of massively-parametric composable function approximators in the form of deep neural networks; cellular solids provide a natural physical analog and—as we show in this work—can also be learned with automatic differentiation.

To make progress towards the goal of learnable morphological computation, in this paper we combine metamaterials with deep neural networks into a framework we refer to as a *neuromechanical autoencoder* (NMA). While traditional mechanical metamaterials are designed for single tasks and actuations, here we propose designs that can solve problems drawn from a *distribution over tasks*, using a neural network to determine the appropriate actuations. The neural network "encoder" consumes a representation of the task—in this case, achieving a particular deformation—and nonlinearly transforms this into a set of linear actuations which play the role of the latent encoding. These actuations then displace the boundaries of the mechanical metamaterial inducing another nonlinear transformation due to the complex learned geometry of the pores; the resulting deformation corresponds to the "decoder". By using a differentiable simulator of cellular solids we are able to learn in an end-to-end way both the neural network parameters and the pore shapes so that they can work in tandem. The resulting system exhibits morphological computation in that it learns to split the processing task across the neural network and the physical mechanism.

The paper is structured as follows. We first introduce the abstract setup for the neuromechanical autoencoder, followed by a brief description of our mechanics model, geometry representation, and differentiable simulation. Although important for the success of our method, the details of our dis-

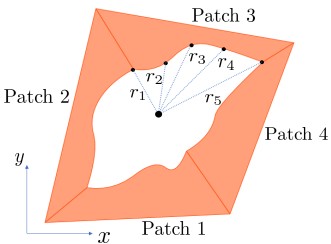
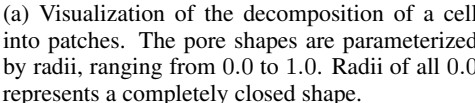

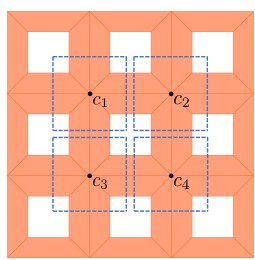

(a) Visualization of the decomposition of a cell into patches. The pore shapes are parameterized by radii, ranging from 0.0 to 1.0. Radii of all 0.0 represents a completely closed shape.

(b) Visualization of a sample initial geometry. The corners of the cells are a geometic parameter; their constraints during NMA optimization are specified by the dotted boxes.

Figure 2: Geometry representation of pores and corners.

cretization and solver for computational nonlinear elasticity problems are in the appendix. We then describe and detail results of our experiments, which include mechanical tasks, a shape matching experiment, and a new mechanical twist on MNIST classification. We end with related work and a discussion on future steps.

## 2 METHODS

### 2.1 NEUROMECHANICAL AUTOENCODER SETUP

We describe the overall setup as pictured in Figure 1. We begin by considering a bounded domain $\mathcal{D} \subset \mathbb{R}^2$ (often square) on which our material exists (in this work we only consider the actuation of 2D geometries). When actuations are applied on the material, its deformation can be described by a displacement field $u : \mathcal{D} \to \mathbb{R}^2$, where $u(\mathbf{X})$ represents the displacement of the particle originally at coordinate $\mathbf{X} \in \mathcal{D}$. For simplicity, assume $u$ can be discretized and identified by a finite-dimensional vector $\mathbf{q} \in \mathbb{R}^N$. The exact form of the discretization is based on a finite element method variant and is described in the appendix in Section A.1.

Next we specify a distribution of tasks $\mathcal{T}$ and an associated loss function $\mathcal{L}(\mathbf{q}; t_i) : \mathbb{R}^N \times \mathbb{R}^m \to \mathbb{R}$, which depends on a *task descriptor* $t_i \sim \mathcal{T}, t_i \in \mathbb{R}^m$, and a displacement field specified by $\mathbf{q}$. The loss function often only looks at the deformation of a subset of the material, such as the displacement of a single point, but we are not restricted to this. The task descriptor is meant to be generic: it can be a coordinate, an image, a scalar parameter, etc.

To map task descriptors to displacements, we use a neural encoder $E_\theta : \mathbb{R}^m \to \mathbb{R}^k$ and a mechanical decoder $D_\phi : \mathbb{R}^k \to \mathbb{R}^N$. The output of the encoder at $t_i$ is understood to be the latent dimension of the autoencoder, and represents the actuations to the mechanical structure. The goal is to choose parameters $\{\theta, \phi\}$ to minimize the loss over the distribution of tasks:

$$\theta^*, \phi^* = \arg\min_{\theta, \phi} \mathbb{E}_{t \sim \mathcal{T}} \left[ \mathcal{L}(D_\phi(E_\theta(t)); t) \right].$$

Given $\nabla_\phi \mathcal{L}(D_\phi(E_\theta(t)); t)$ and $\nabla_\theta \mathcal{L}(D_\phi(E_\theta(t)); t)$ for $t \sim \mathcal{T}$, we can optimize the objective with standard first-order stochastic gradient methods. One difficulty is that $D_\phi(\cdot)$ is an implicit function of its inputs, computed by solving a partial differential equation (PDE). Furthermore, $\phi$ represents geometric parameters defining the domain on which the PDE is solved. To effectively compute derivatives of $D_\phi$, we developed a JAX-based (Bradbury et al., 2018) differentiable elasticity simulator, as described in the next section.

### 2.2 DIFFERENTIABLE SIMULATION

We developed a custom solver for static nonlinear elasticity problems which model the equilibrium of elastic materials under load. The goal is to have a robust and end-to-end differentiable simulator for 2D neuromechanical autoencoders based on mechanical metamaterials. Given geometric design parameters, our solver simulates the structure described by the parameters and computes the gradient (adjoint) with respect to both geometric design parameters and boundary conditions (actuations).

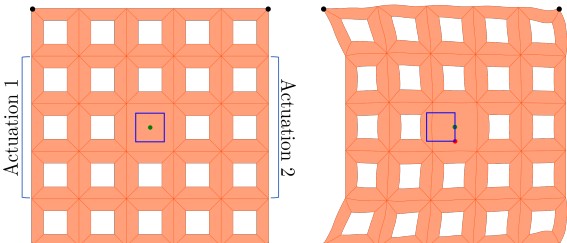

Figure 3: Visualization of the translation task, and the unoptimized initial geometry. The goal is to translate the green pointer to anywhere within the blue square (such as the labeled red dot on the right), using only linear actuations on the two sides. With the starting geometry, only horizontal translations are possible.

In order to make the solver differentiable with respect to geometric parameters, we implement a version of isogeometric analysis (IGA) (Hughes et al., 2005), a finite element method (FEM) (Hughes, 2012) variant where both the underlying solution and geometry basis are based on B-splines. Using B-spline patches allows us to parameterize our geometry in a flexible and yet robust way while maintaining a differentiable map from geometry parameters to PDE solution.

As our simulator is implemented entirely in JAX, we backpropagate gradients directly through both the simulator and a neural network using automatic differentiation and adjoint methods in tandem. In the next sections, we describe the relevant physics and the geometric representation we used.

## 2.3 MECHANICAL MODEL

We give a high level description of the mechanical model here, and detail it further in the appendix. In the static equilibrium problems we consider, the solution is a displacement that minimizes some energy. The elastic properties are captured by a hyperelastic strain energy density function, which depends on the local deformation of the material and is independent of the path of deformation. For a given deformation, the potential energy functional $\Psi(u)$ is the integral of the strain energy density over the material domain $\mathcal{D}$. Given boundary conditions, the resulting physical deformation $u : \mathcal{D} \to \mathbb{R}^2$ is one that minimizes $\Psi(u)$ subject to boundary conditions:

$$u^* = \arg\min_{u \in \mathcal{H}} \Psi(u)$$

where $\mathcal{H}$ is the set of all displacement fields that satisfy prescribed *Dirichlet* boundary conditions (expressed as equality constraints on the displacement field). To solve this in practice, we discretize $u$ and define a standard representation of the geometry. NMA training is bi-level, where in the inner loop we perform the energy minimization using second-order methods. In the outer loop, the solution $u^*$ can be regarded as an implicit function of the design parameters and boundary conditions, and gradients with respect to these can be computed using implicit differentiation.

## 2.4 GEOMETRY REPRESENTATION

The central unit of the metamaterials we design is the cell, a porous shape with a quadrilateral boundary. We initialize the geometry to a regular grid of square cells with simple square pore shapes, similar to that in Figure 2b. During training of the neuromechanical autoencoder, we modify this geometry to minimize the expected loss over a distribution of tasks.

To represent the geometry, we decompose the domain into B-spline patches, each with its own B-spline control points. Each cell is generally composed of four patches; we visualize the decomposition of a representative cell in Figure 2a. The shape of the cell pore is defined by radii (illustrated by $r_i$), whose values specify relative distance of the pore edge from the centroid of the cell (e.g., a cell having radii all $0.0$ corresponds to a completely closed cell). We combine all the radii in all cells into a radii array $r \in [0, 1]^R$, which becomes one of our geometric parameters. For further flexibility, we also allow the shapes of the cells to change within a grid of cells. The corners of the cells, labeled $c_i$ in Figure 2b, are allowed to deviate within a specific box around its values in the initial square lattice-like geometry. Figure 2a shows a cell that its corners perturbed during training. The deviation bound ensures that the shapes do not degenerate during NMA training. The array of corner locations $c$ and radii $r$ comprise our geometric parameters. The outer boundary of the structure is constrained not to change during NMA optimization, as this would otherwise create inconsistent boundary conditions between designs.

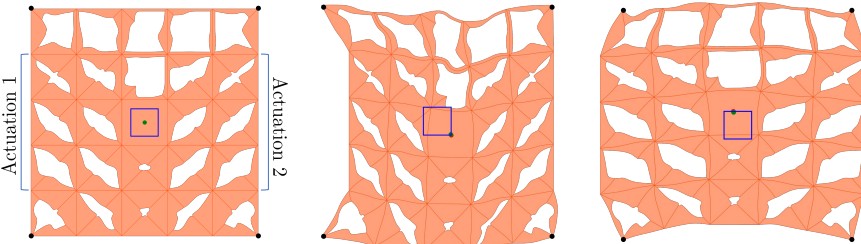

Figure 4: After optimization, we are able to achieve both horizontal and vertical translation using linear actuations. The learned diagonal pore shapes allow compressive actuation to be translated into downwards motion, and tensile actuation into upwards motion. The neural network, jointly learned along with the geometry, successfully translates the goal coordinate to the appropriate actuations.

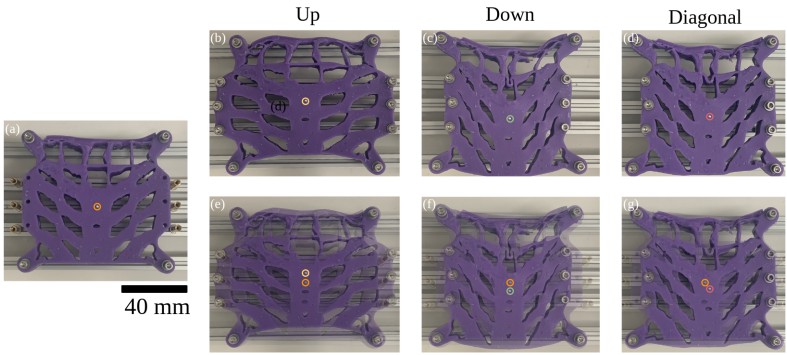

Figure 5: Real-world design of the translation task. (a) the material in its starting unactuated configuration, (b-d) shows the structure in tension, compression, and one sided compression to achieve upwards, downwards, and diagonal pointer displacement. In (e-g) the reference picture (a) is overlayed on top of the actuated material(b-d) to demonstrate the displacement of the pointer.

## 2.5 DISCRETIZATION AND END-TO-END DIFFERENTIABILITY

The details of our discretization are in the Appendix. We mention two important notes here. The first is that careful selection of geometric parameters is critical to being able to differentiate with respect to them. In particular, given the geometric parameters we can construct a differentiable map to the B-spline control points representing the geometry of the model. The analogy in standard FEM would be that our "meshing" operation is fully differentiable. Part of the reason differentiability is always satisfied is that the cardinality and topology of the control points remain the same given any valid setting of geometric parameters. Another important note is that all of our geometric parameters, $\{r, c\}$, are only constrained by simple box constraints, so that first order constrained optimization with them is straightforward. This parameterization is robust in the sense that for any value of the geometric parameters within the box constraints, we have a valid geometry.

## 3 EXPERIMENTS

### 3.1 TRANSLATION AND ROTATION

The first task we tackle is how to perform translation given a limited degree of control. Consider the setup in Figure 3. We have a $5 \times 5$ cellular solid fixed on the corners. The goal is to be able to move the green pointer in the middle of the solid to anywhere within the blue square given only horizontal displacements of the edges. The space of tasks is a small box $\mathcal{B}$, and the task is defined by a single coordinate; the task descriptor $t \in \mathbb{R}^2$ is sampled uniformly from $\mathcal{B}$. The task descriptor is mapped to two horizontal actuations by a simple fully-connected neural network (NN). The loss function is mean squared error of the green pointer after deformation from the goal.

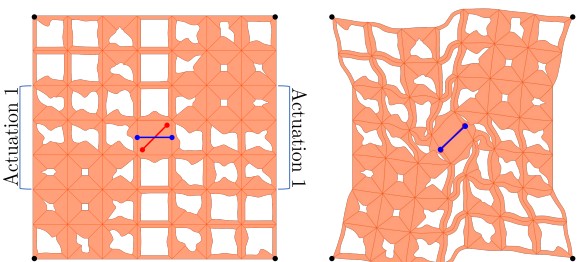

Figure 6: Single direction rotation with a single actuation. The left figure describes the setup and learned geometry. Using a single actuation applied equally on both sides of the metamaterial, we can achieve any single direction rotation up to an angle $\pi/4$ of the middle box.

In the absence of a domain-specific design, the allowed actuations limit the achievable motions to only left-right displacements. As demonstrated in Figure 3 (right), given a red goal point on the bottom right the best we can do with the unoptimized geometry is match the x-coordinate. If we do end-to-end NMA training, however, we converge on the shape in Figure 4. The learned geometry is able to achieve any translation within the blue square. The diagonal pore shapes enable translating compression of the material to downwards motion of the pointer, and tension to upwards motion. Here the joint learning of geometry and control offers a clear benefit: we converge to a nontrivial solution that discovers how to use its geometric nonlinearity with its NN controller.

To demonstrate transfer to the real world, we manufactured the resulting structure and qualitatively verified its behavior. As predicted by our simulation the neuromechanical autoencoder can facilitate displacement of a central point in the upwards via tension, downwards via compression, and diagonally via partial compression. See Figure 5 for details.

The next task is another mechanical task: rotation. We would like to see how the NN and metamaterial can work together to learn to translate linear actuation into rotation of part of the structure. The task description is now the angle of rotation: $t \in [-\pi, \pi]$. We have two setups for this problem. In the first, we use a small fully-connected NN to map angle into a single actuation, applied equally on both sides of the metamaterial (Figure 6). We apply actuations on both sides to avoid translating the middle square in addition to rotation. In the second, the NN maps into two actuations, one applied left-right and one applied top-down (Figure 7). In both cases we consider a $7 \times 7$ metamaterial. The goal is rotation of the blue stick counter-clockwise by an angle $t$ around the center; the loss function is mean square error from the goal of the two points on the ends of the stick.

In the first setup, we are able to achieve unidirectional rotation between $[0, \pi/4]$ (Figure 6, right), while for the second setup, we can achieve bi-directional rotations between $[-\pi/6, \pi/6]$ (Figure 7). Without geometry and control co-design, we would not be able to achieve rotation with linear actuation without an intuition-driven design, but the joint NMA training is able to make good progress.

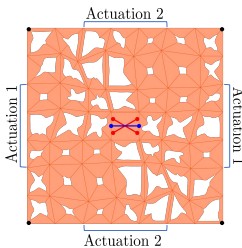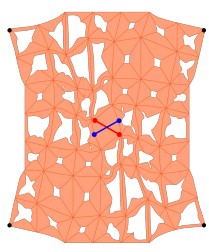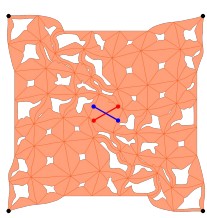

Figure 7: With two independent actuations, we can achieve bi-directional rotation from $[-\pi/6, \pi/6]$. One actuation is applied equally on top and bottom, and the other applied equally left to right.

## 3.2 SHAPE MATCHING

Next we consider a much higher-dimensional task space. Given a family of shapes parameterized by 2-dimensional coordinates, we would like to design a mechanical decoder and a neural encoder that can map coordinates to actuations deforming the structure to resemble a sampled shape from the family as closely as possible.

In particular, we consider a family generated by a log Gaussian process in polar coordinates, approximated via random Fourier features (Rahimi & Recht, 2007). Given a metamaterial with a large central pore, such as in Figure 8a, we would like to deform it to match any of the shapes in the

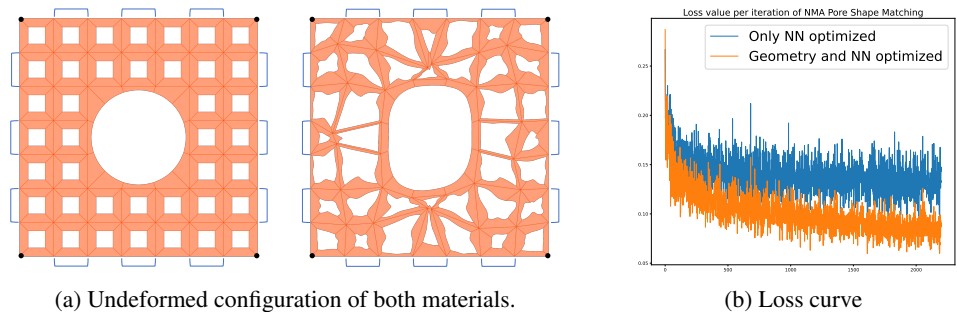

(a) Undeformed configuration of both materials.    (b) Loss curve

Figure 8: Pore shape matching experiments. (a) The undeformed configuration of the unoptimized geometry and the optimized geometry. The 12 actuations applied on the material are depicted with the square brackets. (b) The stochastic loss during training. The optimized geometry achieves a significantly smaller loss since it is able to capture the finer features in the shapes. The geometry can tune itself to the particular random shapes that comprise the dataset.

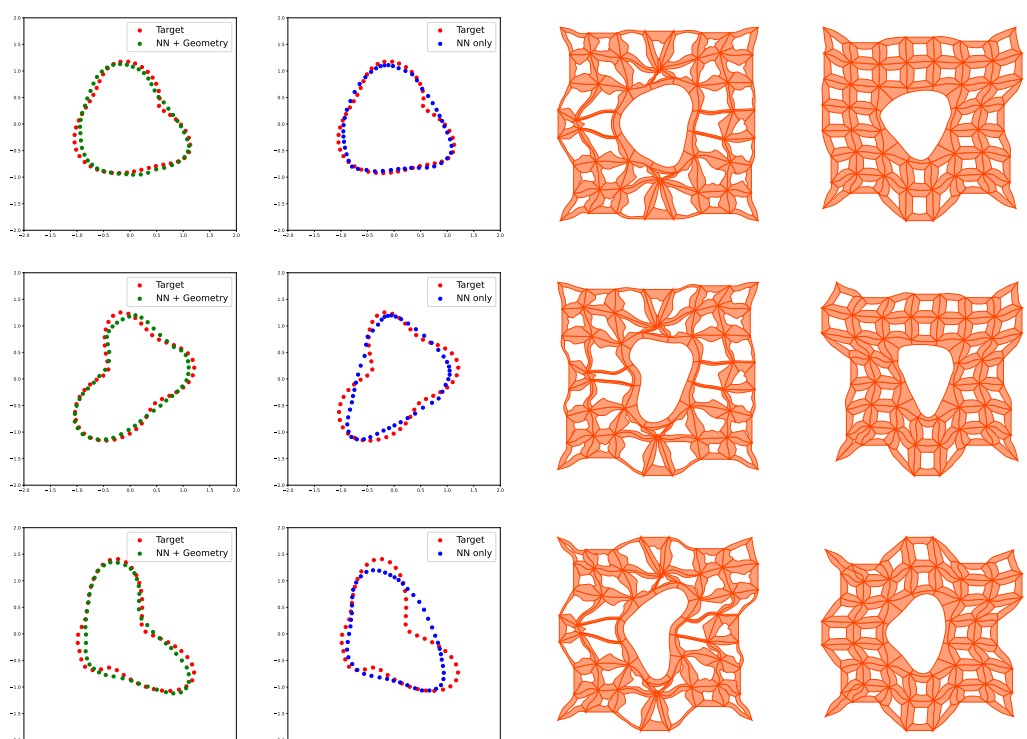

Figure 9: Comparison on random shapes. The optimized geometry is able to capture finer features of the dataset compared to the non-optimized geometry. Note that our loss function is rotation-invariant, so the orientation of the shapes do not necessarily match the orientation of the pores.

family. The task description $t$ is an $n \times 2$ dimensional array of coordinates defining the shape. A fully-connected neural network translates these to 12 actuations applied around the material (as shown in Figure 8a). The final loss function is an $\ell_1$ loss between the control points defining the middle pore after deformation and the points defining the shape. When comparing, we normalize the scale of both shapes, and perform Procrustes analysis for rotation invariance.

We train one version where the geometry and neural network are optimized jointly, and one version where only the neural network is optimized for the starting geometry. Figure 8a shows the non-optimized and optimized geometry. After learning, Figure 9 shows qualitative results of how well the jointly learned metamaterial compares with the control-only material. Jointly learning geometry for the shape family allows us to capture much finer features in the target shape. In Figure 8b

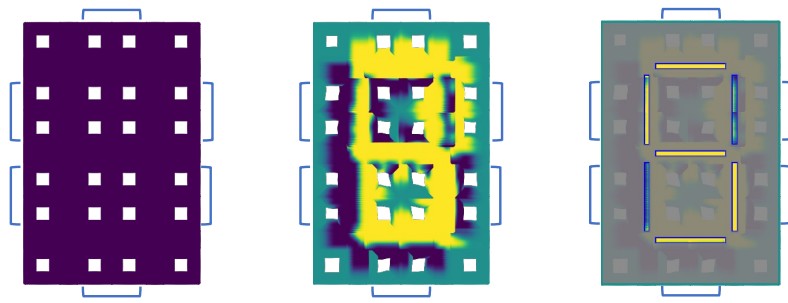

(a) The starting color map.     (b) The optimized color map.     (c) Color map with slits.

Figure 10: The shapes for digital MNIST, along with labeled locations for actuations. The learned color map is highly nontrivial, and allows us to perform underactuated control of the seven slits. Note that the undeformed configuration happens to look like the digit "5", which is surprising as the average of digits across the seven-segment display is not a 5.

we visualize the (stochastic) loss during training. The jointly learned metamaterial converges to a significantly lower loss value, showing the benefit of harnessing the geometric nonlinearity.

## 3.3 DIGITAL MNIST

For our last task we attempt to create a mechanical seven-segment display for classifying MNIST digits. Towards this we add an additional design variable for the material: color. Our starting metamaterial is pictured in Figure 10a, a version of metamaterial that is originally assigned a color value 0 everywhere. We treat color, parameterized by a B-spline patch over the metamaterial, as an additional geometric design parameter that can be optimized with NMA training.

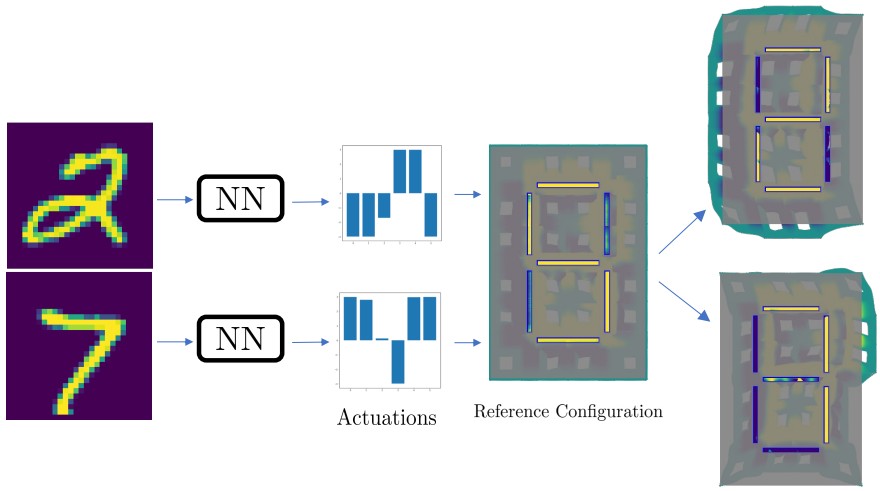

Actuations     Reference Configuration

Deformed Configuration

Figure 11: The setup for the digital MNIST task. The MNIST digit is fed into a neural network, which then produces actuations to deform the material in six locations. After deformation the slits show a digital seven-segment representation of the MNIST digit.

Our input to the neural network is an image sampled from the MNIST dataset. The neural network then produces actuations that deform the metamaterial to produce a seven-segment representation of the MNIST digit when viewed through small slits. Figure 10b visualizes the learned colop map, and Figure 10c shows the structure with slits added. The loss function is manually specified for each digit, e.g. if an MNIST digit has a label of "1" then the right two slits should contain color value 1.0, while the rest should contain 0.0. The full setup is displayed in Figure 11, with samples after training displayed in Figure 12. Although this can be learned from scratch end-to-end, to speed up training we first learned colors and actuations to be able to reproduce all 10 digits, and then trained

a small feed-forward neural network to match the actuations for each digit. We then set up the entire pipeline and finetuned end-to-end for better performance. We note that the 7 segments are controlled by only 6 actuations, so by restricting the family of objects displayed to the 10 digits, we allow the neural encoder and mechanical decoder to learn underactuated control of all 7 segments. Additional samples are presented in the Appendix. We also note that the pore shapes did not have to change significantly to accomplish this task. The only "geometry" design was through the coloring, which as visualized in Figure 10b turns out to be highly nontrivial.

## 4 DISCUSSION AND RELATED WORK

**Differentiable Simulation**  The abundance of differentiable simulators has demonstrated their usefulness in designing novel systems.  Hu et al. (2019) developed a differentiable simulator with hand-written custom CUDA physics kernels that enabled material inference, control of a soft walker, and co-design of a swinging robot arm.  Sanchez-Gonzalez et al. (2020) presents an ML framework to model a variety of physical domains to solve forward and inverse problems using a graph neural network approach. Mozaffar & Cao (2021) developed a differentiable finite element simulator to control and infer material parameters within the context of additive manufacturing processes. Liang et al. (2019) developed a differentiable cloth simulator, and Ham et al. (2019) automated the calculation of weak shape derivatives within the context of finite elements to solve PDE constrained shape optimization problems.  In our work, our differentiable simulator is developed specifically to aid in neuromechanical autoencoder design.

**Mechanical Metamaterials**  As the rational design of nonlinear mechanical materials is often unintuitive, modern machine learning approaches have enabled faster design.  Deng et al. (2022) coupled a neural accelerated mass spring model that facilitated an evolutionary approach to design functional structures.  Mao et al. (2020) applied generative adversarial networks to design unit cells for architected metamaterials, Kumar et al. (2020) introduces a novel class of anisotropic meta-

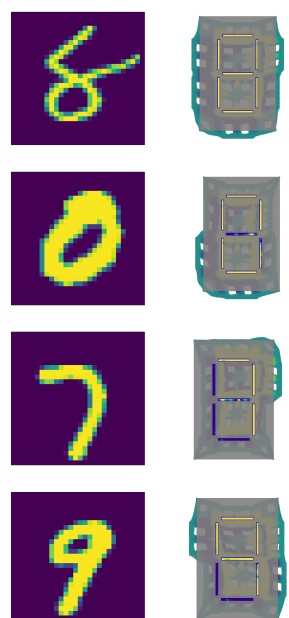

Figure 12: Some MNIST samples and results.  Additional samples available in appendix.

materials and a machine learning method for the inverse design of their geometry given desired elasticity properties, Beatson et al. (2020) learned a reduced order model to speed up simulation of cellular metamaterials, Xue et al. (2020) introduced a homogenization approach for cellular metamaterials, and Xue & Mao (2022) introduced a mapped shape approach to design metamaterials to fit a prescribed strain energy curve.  Our work uses classical gradient/adjoint methods to optimize the geometric parameters, but could be combined with machine learning methods to speed up simulation and hence faster NMA training.

### 4.1 LIMITATIONS AND FUTURE WORK

We introduce the framework of neuromechanical autoencoders, inspired by the biological co-evolution of control and morphology. We present a method for automatic design of these systems, and show a number of results that produce nontrivial behavior through co-design, both in simulation and in real-world.  We believe this is a small but significant step in the road to designing mechanically-intelligent systems. The two major bottlenecks in our approach is the runtime of PDE solving and geometry parameterization. Fast PDE solving, especially for nonlinear PDEs such as the ones we use, is a very active area of research, and is crucial to scaling up NMA design. In terms of geometry parameterization, the key is to find a space of materials that have a complex range of mechanical deformation properties and yet are easy to simulate. For this paper, 2D cellular solids with nonuniform pore shapes were a great ansatz, but a future work could understand and quantify how much "computation" these materials can do. Scaling up to 3-dimensional intelligent mechanical models, as well as including dynamics, would significantly improve the computation capabilities, but would require much faster solvers. This is a key focus in our further work.

## 4.2 ACKNOWLEDGEMENTS

We would like to thank Alex Beatson, Geoffrey Roeder, Jordan Ash, and Tianju Xue for early conversations around this work. We also thank PT Brun for assisting with fabrication. This work was partially supported by NSF grants IIS-2007278 and OAC-2118201, the NSF under grant number 2127309 to the Computing Research Association for the CIFellows 2021 Project, and a Siemens PhD fellowship.

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

# A  APPENDIX

## A.1  DISCRETIZATION AND IGA

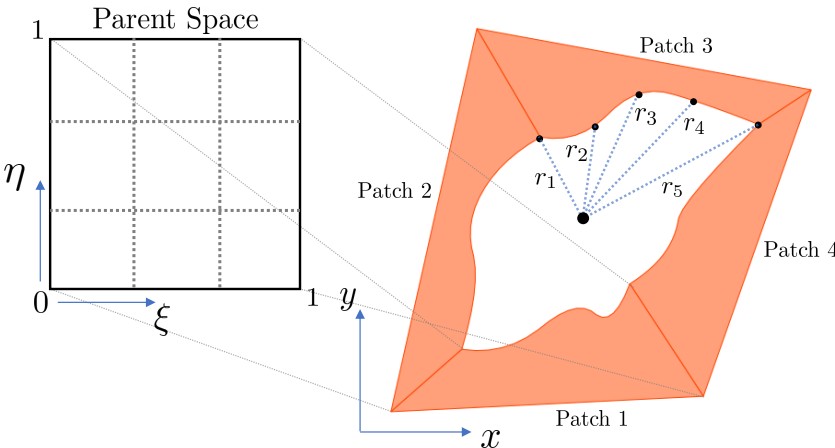

Figure 13: Visualization of the decomposition of a cell into patches. Each patch is pulled back into a parent space, which is a B-spline knot span. Quadrature and integration happens in this space. The pore shapes are parameterized by radii.

We discretize the geometric domain into $P$ isogeometric patches. All patches use the same B-spline basis functions $B^{ij}(\xi, \eta)$ Piegl & Tiller (1997) with $i, j \in [n_p]$ and $\xi, \eta \in [0, 1]$, where $n_p$ is the number of control points. The domain of $\xi$ and $\eta$ correspond to the *parent domain* of each patch (Figure 13). In the B-spline literature, the parent domain is often referred to as the knot span. The basis functions are piecewise polynomial with degree specified as a parameter. In all of our experiments, we use piecewise quadratic B-spline functions. Each B-spline basis function corresponds to a control point, and each control point represents two degrees of freedom in 2D space, i.e., each patch $p \in [P]$ has $n_p \times n_p \times 2$ degrees of freedom ($x_p^{ij}$ and $y_p^{ij}$).

The mapping from the parent domain of each patch to the physical domain is given by a linear combination of B-spline basis functions, where the weights of the linear combination are given by the control point coordinates. Explicitly, the mapping function $\phi_p$ from parent space of patch $p$ to physical space is given by

$$\phi_p(\xi, \eta; \boldsymbol{x}, \boldsymbol{y}) = \sum_{i=1}^{n_p} \sum_{j=1}^{n_p} \begin{bmatrix} x_p^{ij} \\ y_p^{ij} \end{bmatrix} B^{ij}(\xi, \eta), \tag{1}$$

where $[x_p^{ij}, y_p^{ij}]$ are the control points parameterizing the mapping. In our simulator, we represent the reference configuration with reference control points $[X_p^{ij}, Y_p^{ij}]$; these are determined by our geometry parameters $\{r, c\}$ through a differentiable map. We then parameterize the deformed geometry with the same basis, using deformed control points $[x_p^{ij}, y_p^{ij}]$. For a given deformation, the integral in Eq 3 representing the potential energy can be computed by a pullback in the parent domain, using standard Gaussian quadrature as is standard in FEM (Hughes et al., 2005).

Dirichlet boundary conditions of the type we use in this paper can be represented as constraints on a subset of the control points. For each boundary condition, the corresponding reference and deformed control points are prescribed to have particular displacement values. Furthermore, since our geometry is decomposed into multiple neighboring patches, the control points must also have incidence constraints amongst them. These are kept track of using constraint groups, where each group has a representative element.

## A.2  MECHANICAL MODEL

In the static equilibrium problems we consider, the solution is a displacement that minimizes some energy function. In particular, we use a nearly incompressible Neo-Hookean material model (Ogden,

1997), in which the elastic properties are captured by a hyperelastic strain energy density function. This function, $W(\mathbf{F})$, $W : \mathbb{R}^{2 \times 2} \to \mathbb{R}$, is independent of the path of deformation and is a function of the deformation gradient tensor, $F_{ij} = \partial u_i / \partial X_j + I$ where $\mathbf{X} \in \mathcal{D} \subset \mathbb{R}^2$ represents the position of a particle in the undeformed *reference* configuration, and $u(\mathbf{X})$ is the displacement field. Here,

$$W(\mathbf{F}) = \frac{\mu}{2}(I_1 - 2 - 2 \log J) + \frac{\kappa}{2}(\log J)^2, \qquad (2)$$

where $J = \det(\mathbf{F})$, $I_1 = \text{tr}(\mathbf{F}^T \mathbf{F})$, and $\mu = E/2(1+\nu)$ and $\kappa = E/3(1-2\nu)$ are shear and bulk moduli of a material with Young's modulus $E$ and Poisson's ratio $\nu$, respectively. This is a standard choice for hyperelastic material modeling that transfers well to the real-world. We can solve for the displacement by finding the stationary point of the potential energy functional $\Psi(u)$,

$$u^* = \arg \min_{u \in \mathcal{H}} \Psi(u) \qquad \Psi(u) = \int_{\mathcal{D}} W(\mathbf{F}) d\mathbf{X} = \int_{\mathcal{D}} W \left( \left. \frac{\partial u}{\partial \mathbf{X}} \right|_{\mathbf{X}=\mathbf{X}'} + I \right) d\mathbf{X}' \qquad (3)$$

where $\mathcal{H}$ is the set of all displacement fields that satisfy prescribed *Dirichlet* boundary conditions (expressed as equality constraints on the displacement field). To solve this in practice, we discretize $u$ and define a standard representation of the geometry. Abstractly, the solution $u^*$ can be regarded as an implicit function of the design parameters and boundary conditions. The solution $u^*$ can be computed in a discretized form using standard second-order optimization algorithms, and gradients can be computed using implicit differentiation.

## A.3    END-TO-END DIFFERENTIABILITY

We would like to reiterate that our differentiability conditions are satisfied, so that we can train neuromechanical autoencoders end-to-end. We first define a differentiable map from geometry parameters and Dirichlet boundary condition values into the reference B-spline control points. This is then used to construct the $\Pi_l, \Pi_g$ functions, both of which are differentiable. The global vector $\mathbf{q}$ is then passed into a black-box optimizer to produce the solution $q^*$. Gradients with respect to the solution of the optimizer are computed using adjoint optimization. The solution $q^*$ is then mapped back into local coordinates using $\Pi_l$, and is used to compute the loss function $\mathcal{L}$.

This pipeline ensures that we have a differentiable map from geometry parameters and boundary conditions (actuations) to the NMA task loss function, so we can proceed to train the NMA objective using stochastic gradient descent. In the next section, we demonstrate specific applications.

## A.4    SOLVER DETAILS

After discretization to $q$, we solve the energy minimization using Newton's method with incremental loading. The Hessian of the energy is assembled in sparse form using the trick from Powell & Toint (1979). Using the discretization of the system we automatically derive the sparsity pattern of the Hessian, and then construct appropriate binary vectors to perform Hessian-vector products with. We then reshape these into a CSR matrix representation of the Hessian.

The sparse linear systems are then solved by GMRES (Saad & Schultz, 1986) preconditioned by an incomplete LU decomposition. Since the energy involves $\log \det F$, where $F$ is the deformation gradient, taking a finite step can lead to numerical blowup. Therefore our incremental loading is adaptive, and a line search is performed to avoid inversion of elements in the geometry.

## A.5    EXPERIMENTAL DETAILS

All B-spline patches used were quadratic and contained $5 \times 5$ control points. Quadrature was done by degree 5 Gauss-Legendre. Most computation was done on NVIDIA RTX 2080 GPUs. Each solve instance was done on a single GPU, and mini-batching was done by parallelizing with MPI. Each MPI task used a single GPU. The radii parameters were clipped to $[0.1, 0.9]$ to aid solver stability.

### A.5.1    TRANSLATION TASK

The learning rate was $0.0001 * M$ where $M$ is the number of MPI tasks. In this case, we used 8 MPI tasks. The neural network was a fully-connected network with activation sizes: $2 - 30 - 30 - 10 - 2$ (including input/output). The final layer was clipped by a $\texttt{tanh}$ and multiplied by a maximum displacement of 60% of cell width.

### A.5.2 ROTATION TASK

The learning rate was $0.001 * M$ for single and $0.01 * M$ for double rotation. $M$ is the number of MPI tasks. In this case, we used 8 MPI tasks. The neural network was a fully-connected network with activation sizes: $1 - 20 - 10 - 1$ (including input/output) or $1 - 20 - 10 - 2$ for the double actuation. The final layer was clipped by a $\tanh$ and multiplied by a maximum displacement of $60\%$ of cell width.

### A.5.3 SHAPE MATCHING TASK

The learning rate was $0.0001 * M$ where $M$ is the number of MPI tasks. In this case, we used 16 MPI tasks. The neural network was a fully-connected network with activation sizes: $98 - 200 - 200 - 12$ (including input/output). The final layer was clipped by a $\tanh$ and multiplied by a maximum displacement of $60\%$ of cell width.

### A.6 DIGITAL MNIST TASK

Initially we learned colors and actuations for a lookup table of 10 digits. This was trained with a learning rate of $0.01 * M$ where $M$ is the number of MPI tasks. We used 10 MPI tasks, one per digit. We clipped the maximum displacement to $60\%$ of cell width using $\tanh$. Afterwards, we trained a fully-connected neural network to map MNIST digits to the actuations of the corresponding digit. We then put the neural network to map directly to actuations, and finetuned end-to-end with a learning rate of $0.0001$.

## A.7   ADDITIONAL PORE MATCHING RESULTS

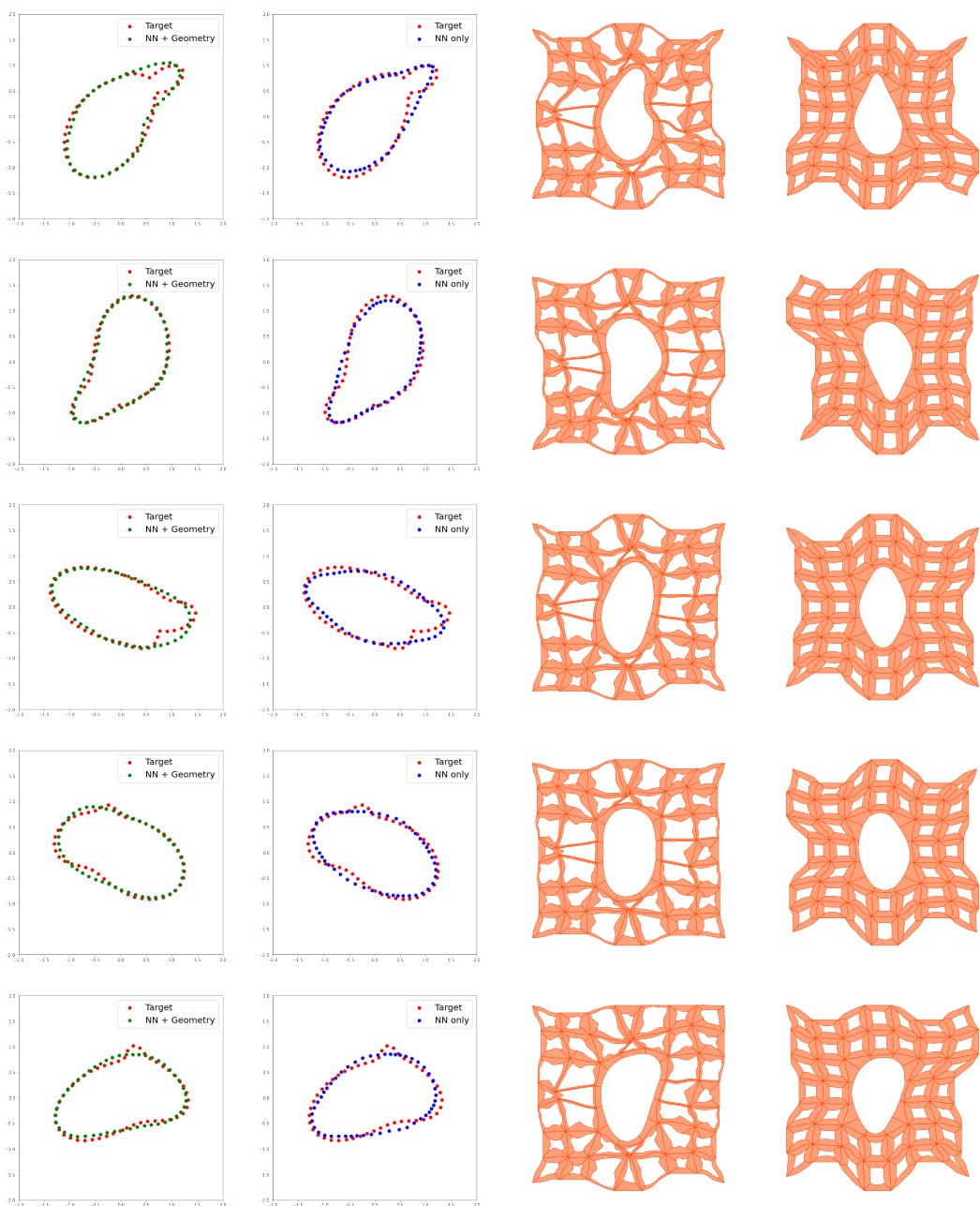

## A.8 Additional Digital MNIST Results

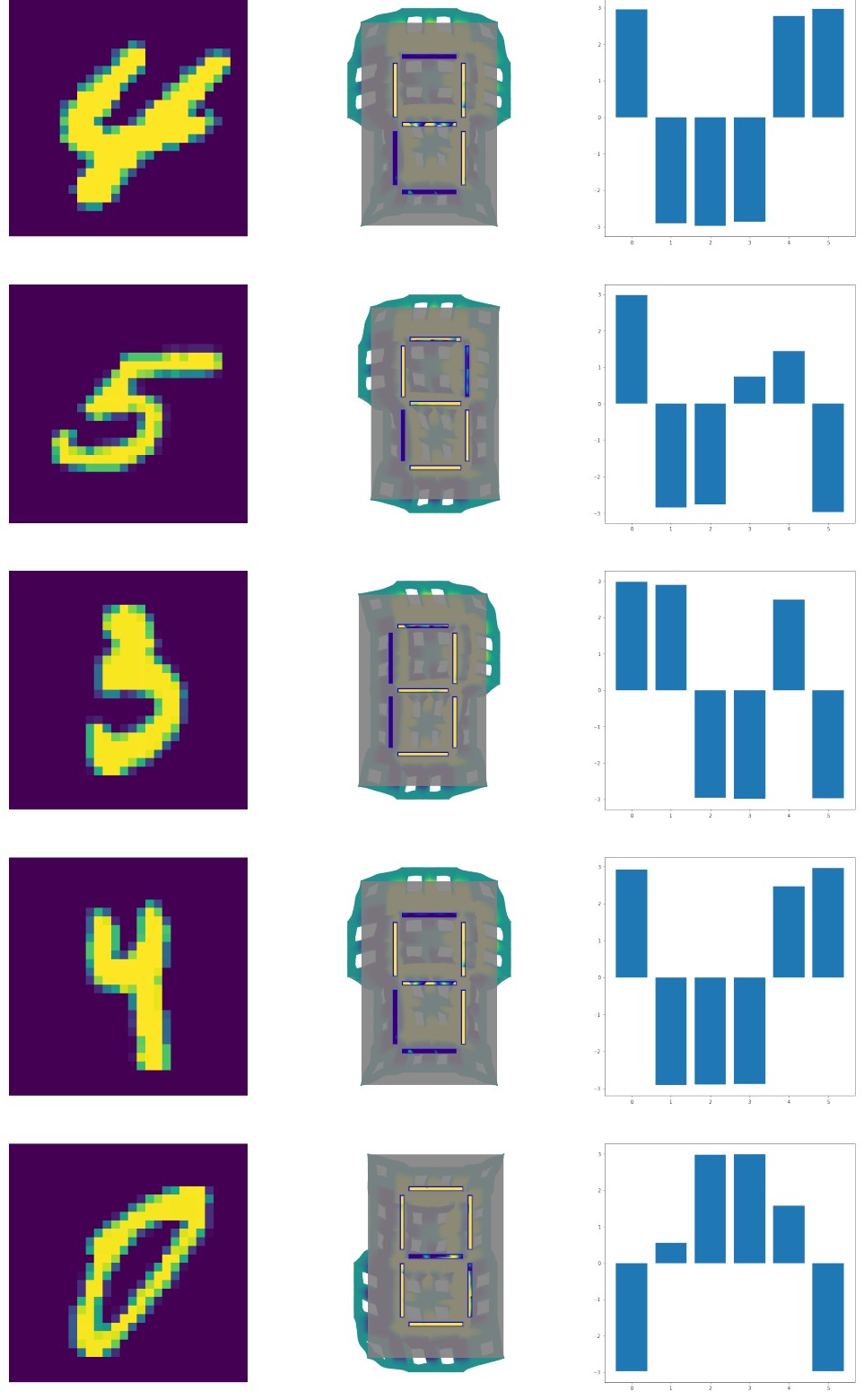

