# OpenReview forum: "Neuromechanical Autoencoders: Learning to Couple Elastic and Neural Network Nonlinearity"
_ICLR.cc/2023/Conference — ICLR 2023 notable top 25%_

### Official Review · Reviewer_fAWH · 2022-10-17

**Confidence:** 3
**Correctness:** 4
**Technical Novelty And Significance:** 4
**Empirical Novelty And Significance:** 4
**Recommendation:** 8

**Clarity, Quality, Novelty And Reproducibility:**

The paper is well written throughout. Just a  few clarifications:

- What exactly are the "actuations" mentioned in the figures? We just see blue vertical or horizontal brackets on each side of the square, but it's not clear what the actuation actually does - I initially thought  it would be squeezing along the direction of the bracket but that doesn't seem to be the case.

- The purpose of section 2.3 might be explained a little  bit more - what exactly are we computing here and how is it used in the system?

- Caption of Figure 8 a): should the first "optimized" be "unoptimized"?

- Typo in the last paragraph of p. 4, "e.g.,wel"


**Strength And Weaknesses:**

Strength:

- The method is novel (to my knowledge) and interesting. While there has been much work on optimizing morphologies and controllers jointly (e.g. Karl Sims and other "virtual creatures" models - these should be cited!), as well as optimizing elastic shapes (e.g.  Sodarace), the system described here  looks like nothing I'm aware of.

Weaknesses:

- The only minor defect I can see is that some points in the explanation could be clarified - see below

**Summary Of The Paper:**

= Update after rebuttal =

I thank the authors for their response to my comments. I maintain my recommendation for acceptance.

= Original review =

The authors show how to optimize the shape of an elastic material, as well as neural networks to control its actuators, imn order  to make the material perform various mechanical tasks.

Both the material's  shape and the controlling neural network  are optimized by gradient descent for various task domains. Given a target action (which,  depending on the  domain, might be a location for  a point, a shape,  or even an MNIST digit image to be classified), the neural network translates a specification of  the action into adequate actuations, through which the optimized shape performs the  specified action.

The authors apply their method to translation/rotation of a point on the material, controlled deformation of a  shape, and -  remarkably - translating MNIST images into seven-segment digit displays through deformation of a colored shape  seen through slits.

The authors also show a working real-life implementation of the system, demonstrating practical applicability.

**Summary Of The Review:**

It's new, interesting, and seems to work well. Easy accept.

---

> ### Author Response · Authors · 2022-11-07
> **Response to review**
>
> Thank you for reviewing our paper! We are glad you found the work novel and interesting. We also appreciate the pointers to virtual creatures models. We will be sure to cite them.
>
> ### Actuations:
>
> Actually you are right, the actuations are how much to stretch (or push) on the areas designated by the brackets. The neural network outputs a scalar for each actuation, constrained between actuation limits (usually [-3.0, 3.0], which corresponds to a 3/5th of a unit cell). A positive actuation corresponds to a push, and negative corresponds to a pull.
>
> ### Section 2.3:
>
> This is the material model that allows us to model the deformation of a solid. The actuations correspond to the boundary conditions on a nonlinear partial differential equation. The solution of the partial differential equation can be given by an energy minimization problem constrained by the boundary conditions: Find a displacement field that minimizes a potential energy (potential energy determined by the specific material model) while respecting the boundary conditions. This is the “inner loop” in the NMA training scheme. NMA can hence be thought of as some sort of bilevel optimization. The outer loop is decreasing a task objective function, while the inner loop is doing energy minimization to figure out deformation of the mechanical structure under actuation. It seems multiple reviewers have been confused a bit by this section, and we will be revising it.

---

### Official Review · Reviewer_S3up · 2022-10-25

**Confidence:** 3
**Correctness:** 4
**Technical Novelty And Significance:** 4
**Empirical Novelty And Significance:** 4
**Recommendation:** 8

**Clarity, Quality, Novelty And Reproducibility:**

I believe that the presented work is technically sound and makes highly novel contributions with potential applications in interdisciplinary areas. The work has been written clearly, yet, I think it will be hard for the broader audience to grasp some important parts of the paper as mentioned in my review. In terms of reproducibility, the authors have shared their code used in the paper to reproduce results.

**Strength And Weaknesses:**

Strengths:
- The paper presents a highly novel and technically sound integration of neural network-based encoders with differentiable mechanical decoders. This combination potentially has several applications in the real-world such as robotics, material design, 3D printing, etc.
- The digital MNIST task that the authors develop -- in which the neural encoder receives an RGB image of a digit as input and the mechanical decoder produces deformations to the metamaterial that classifies the digit -- is very intriguing and a great demonstration of learning (with gradient descent) a solution to the common and accessible MNIST task through morphological computation.
- The authors share their code as part of the submission, this should is helpful in more thoroughly understanding the proposed work.

Weaknesses:
- I found it hard to grasp Methods subsections 2.3 and 2.5. I am not an expert in this domain and I'm not surprised that I find it hard to read these sections, but it would be great if the authors could please see if they can further improve readability of these sections for the broader audience.

**Summary Of The Paper:**

The authors introduce neuromechanical autoencoders (NMA) -- neural encoder combined with a differentiable mechanical decoder -- to learn to perform morphological computation using gradient descent. The authors show how it is feasible for the decoder to perform translation, rotation and matching shapes specified to the encoder via actuations output by the encoder. They also demonstrate the performance of NMA on a material they manufactured to verify how the model works in the real-world.

**Summary Of The Review:**

I find the presented work to be highly novel and potentially impactful for the intersection of machine learning and multiple basic sciences. I believe this line of work is important and it looks like the domain is very new and understudied. I find this work to be very relevant to representation learning and the ICLR audience, and hence recommend acceptance. That said, I am not an expert in this area and am willing to read other reviews and update my score based on potential weaknesses I may have missed to identify.

---

> ### Author Response · Authors · 2022-11-07
> **Response to review**
>
> Thank you very much for reviewing our paper! We are glad you found the work to be novel and interesting.
>
> ### Technical accessibility:
>
> We agree that there is a lot of technical material here that should be communicated appropriately, especially to the broader machine learning audience whom we would love to get excited about this work. We are thinking about how best to do this. Our current thought is to expand on the techniques significantly in the appendix, while giving a broad overview and providing appropriate pointers in the main text. This has mostly been our approach already, but we will be doing a few more passes on it. Thank you for providing specific sections that were unclear!

---

> > ### Comment · Reviewer_S3up · 2022-11-25
> > **Acknowledgment of authors' response**
> >
> > I thank the authors for their response to comments in my review. I appreciate the authors' efforts in enhancing the accessibility of the proposed work to the broader machine learning audience. After author response, I am retaining my score and still believe that this paper makes a significant contribution that is relevant to ICLR.

---

### Official Review · Reviewer_twBy · 2022-10-25

**Confidence:** 5
**Correctness:** 3
**Technical Novelty And Significance:** 2
**Empirical Novelty And Significance:** 3
**Recommendation:** 6

**Clarity, Quality, Novelty And Reproducibility:**

The paper is well written and a good demonstration of machine learning application. Novelty in the machine learning algorithms is not present. Code is shared for reproducibility

**Strength And Weaknesses:**

Strengths:

* The use of neural networks to converts tasks to actuations and that used along with the simulation-based decoder in a single differentiable framework shows great promise in achieving nontrivial displacements
* The adoption of cellular solids and B-spline patches into the mechanical simulation/decoder seems to work well to produce stable solids.

Weaknesses:
* No new algorithmic/machine learning approaches have been proposed
* The work is just coupling the a simple multi-layer perceptron to a well-designed simulator (PDE solver) to achieve the joint optimization of geometry and boundary conditions
* Time complexity of training/learning in this approach is not discussed
* What is the motivation for using L2 loss in one case and L1 in the other?
* How expensive is the generation of training data; is this approach scalable to train with larger datasets for more complex deformations?
* What is the reason to choose the Neo-Hooken material model?


**Summary Of The Paper:**

This works presents a co-design of metamaterials based mechanical/morphological actuation and neural network-based control. The control tasks are allowed to be image, coordinate on scalar parameter based on the end goal, which are mapped on to a latent dimension (using an encoder) matching the number of linear actuations. The decoder is a differentiable simulation that solves for equilibrium of non-linear elastic materials under mechanical load that takes the geometric design parameters and actuations (output from encoder). The solver is made differentiable wrt the geometry parameters (represented as a porous shape with quadrilateral boundary) by adopting isogeometic/FEM analysis and B-spline basis to solve the PDE. The results show that the geometry and control co-design is able to achieve translation, rotation and shape matching efficiently and better than geometry-only scenario.  It is interesting to see the digital MNIST benchmark and the application of this approach to that case

**Summary Of The Review:**

The geometry and control co-design using differentiable simulator and neural network combination shows good results, but no algorithmic innovations in machine learning are proposed in this work. More details need to be provided about the approach to better assess its applicability for other general scenarios.

---

> ### Author Response · Authors · 2022-11-07
> **Response to review**
>
> Thank you for your review of our paper. We would first and foremost like to address your comment about the lack of “new algorithmic/machine learning approaches”. We have a few points along this line:
> - In the call for papers, ICLR accepts “application” papers in its relevant topics. We believe this is a solid application paper with a novel setup that has demonstrated good, interesting, and unintuitive results across several tasks. Even if it had not introduced new algorithmic machine learning approaches, we believe this is new, impactful, and relevant knowledge.
> - This being said, we do actually believe we have introduced a new algorithmic/machine learning approach. Our framework of jointly learning morphology and neural network control over a distribution of tasks by sampling mini-batches of tasks, training the neural network to translate the task into actuations, and training this all end-to-end using stochastic gradient descent with a (custom designed) differentiable simulator, is to the best of our knowledge novel and we think is an important new algorithmic approach to learning morphological computation.
> - A central defining theme of ICLR is representation learning, and we believe this captures the essence of representation learning very well. The joint training of the neural network and mechanical structure allows them to learn a common representation: the actuations. The neural network takes a human-interpretable task description (a picture, coordinate, rotation angle, etc) and translates it to the representation the mechanical structure understands. This is why we have dubbed the system a Neuromechanical Autoencoder: the latent space of the “autoencoder” is the common representation between the neural network controller and the mechanical structure.
>
> ### Time complexity / scaling of method:
>
> Each iteration of NMA simulation involves solving multiple highly nonlinear partial differential equations in parallel (one for each element in the minibatch), and this is unfortunately a slow operation. The more complex the deformations, the longer the simulation will take. In all the experiments we have in the paper, the experiments take about 10-30 seconds per iteration (iterations can vary also within the minibatch). We do this in parallel via MPI for each task in the sampled minibatch, and each runs on a single GPU independently. We used 8 GPUs for the translation/rotation tasks, 16 for shape matching, and 10 for MNIST. We let each experiment run for about 2000 iterations, so the total time taken is less than 24 hours for each experiment.
>
> There are plenty of ways to speed this up, as fast PDE solving is a very active area of research both in the computational science and graphics literature. Our focus in the paper was not on fast solvers. Our goal was to demonstrate the Neuromechanical Autoencoder framework and show that it can achieve interesting results. Scaling this up to 3D, much larger problems with even more complex physics, is a very interesting next step that we are exploring.
>
> ### How expensive is generating training data:
>
> None of the experiments are bottlenecked by generating training data. For the shape matching task, we sample simple fourier features. The approach could generalize to more complex datasets, but it could require materials that are capable of even more complex deformations. It is a very interesting further research direction to see how we can quantify the “expressivity” of the mechanical structure, in terms of the “computation” it can perform. The key is to find a space of materials that have a complex range of mechanical deformation properties and yet are easy to simulate. For this paper, 2D cellular solids with nonuniform pore shapes were a great ansatz.
>
> In the conclusion of the paper, we will add discussion of these above two issues, as they discuss the potential weaknesses and strongly motivate future work.
>
> ### Motivation for L2 loss vs L1 loss:
>
> We chose the L1 loss in the shape matching task to capture the fine features of the shape more easily. We had initially tried L2 loss, but then realized that the learned shape would find that it is “good enough” to just be close to a circle and not capture the curvature. For the other tasks, L1 or L2 would work equally well, since those are low dimensional (2-4 dimensions to match the translation point or rotation points).
>
> ### Why did we choose Neo-Hookean:
>
> The Neo-Hookean material model is a standard choice among hyperelastic materials to manufacture mechanical metamaterials both in the mechanics and graphics literature. It is a robust nonlinear material model that empirically transfers well to the real world (as we can also confirm in our fabrication experiment).

---

> > ### Comment · Reviewer_twBy · 2022-11-29
> > **Acknowledgement of author's response**
> >
> > I would like to thank the authors for the detailed response. This definitely helps to clear up a lot of my questions. I raised my score based on that.
> >
> > I very much agree that this is a novel application (paper) of the "encoding" and representation learning by learning morphology and neural network control using a distribution of tasks. However, there are other works in the literature that find representations across (distribution of) tasks that are amenable for control (such as meta learning [1]). If the claim is algorithmic novelty then it is warranted to have a discussion/comparison with them.
> >
> >
> > [1] Finn, Chelsea, Pieter Abbeel, and Sergey Levine. "Model-agnostic meta-learning for fast adaptation of deep networks." In International conference on machine learning, pp. 1126-1135. PMLR, 2017.

---

### Official Review · Reviewer_Fw7y · 2022-10-26

**Confidence:** 2
**Correctness:** 4
**Technical Novelty And Significance:** 4
**Empirical Novelty And Significance:** 4
**Recommendation:** 8

**Clarity, Quality, Novelty And Reproducibility:**

Clarity: Extremely high! Section 2.1 is a little difficult to digest at first read.

2.4 2nd paragraph, 2nd brackets ‘wel’?

I am a little surprised at the spiky shapes emerging in many of the shown optimised geometries. Is this an optimisation issue? Or is this some repeating, near optimal, solution (maybe one that also appears in nature after evolution)?

The MNIST experiments are really cool, although I am thinking that the encoder network probably just does the classification and then translates this into the actuation for the 7 slit representation. But even if this is a bit of a gimmick, it is still a very cool demonstration!

Quality: Very high, as far as I can tell. It might be nice to include a section with limitations and optimisation pitfalls, i.e., all the things you tried on the way that did not work – that would be helpful for other researchers to build on this work.

Novelty, I cannot really judge because I don’t work in this area of meta materials. However, for what it is worth, I have never seen a demonstration like this and think that it is really creative and novel.

**Strength And Weaknesses:**

Strength: The paper is very well written, easy to follow and fun to read. Even for a reader unfamiliar with meta materials and the associated physics, they manage to write an engaging scientific exploration. The proposed model is very creative and works at an exciting intersection of material and data science.

Weaknesses: It would have been great to include a tiny bit more detail about technical things, for instance, how is the adjoins method implemented, which PDE solver is used etc. However, I do realise that this might come at the expense of making the paper less accessible.

**Summary Of The Paper:**

I thank the authors for their rebuttal replies and for further improving the quality of this paper. I think my score is still accurate and I look forward to seeing this work published.

The paper proposes a differentiable model for learning meta material layouts and neural network encoder that can produce actuations to control these materials. They show through a number of simulations that this works with simple translations, rotations and complex shapes. They also build this material and show that it works. Finally, they show how this combined framework can be trained to translate MNIST digits into 7 slits representations.

**Summary Of The Review:**

The authors propose a model combining a neural network controller and a learned geometry. This is a very creative solution to a problem in material sciences (I suppose) and it is implemented with some cool technical methods based on spline representations, adjoins computation and extended efforts to build a differentiable system that works under a number of physical and geometric constraint. I think that this is a really project and would be happy to see it published – supposing that other reviewers which are more familiar with this research area agree on its novelty and soundness.

---

> ### Author Response · Authors · 2022-11-07
> **Response to review**
>
> Thank you for your review! We are very glad you enjoyed our paper.
>
> ### More technical details:
>
> We agree that it would be a good idea, and are thinking of the best way to incorporate it. We already have some of the technical details in the appendix, but we want to expand it for those who want to delve more.
> We are looking to improve section 2.1 and 2.3 to make it more accessible, maybe by putting some of the more technical details in the appendix (especially section 2.3).
>
> ### Spiky pore shapes:
>
> Our guess is that this is due to the way we parameterized the pores. The pores are made up of 4 patches, and within each patch the pore shape is represented by a quadratic spline, while between patches we only have C^0 continuity. If we want to enforce higher continuity between patches, we can either constrain the neighboring radii, or add some penalty term to the training objective to avoid spike patterns.
>
> ### Limitations:
>
> We will be adding a discussion on limitations in the conclusion. The main limitations are runtime (solving highly nonlinear PDEs is very difficult) and parameterization of geometry. In terms of geometry parameterization, the key is to find a space of materials that have a complex range of mechanical deformation properties and yet are easy to simulate. For this paper, 2D cellular solids with nonuniform pore shapes were a great ansatz, but a future work could be to understand and quantify how much “computation” these materials can do.

---

### Author Response · Authors · 2022-11-11
**Summary of revision**

Hi all, thank you for the feedback in your reviews. We have uploaded a revision with some small changes:
- Section 2.3 (Mechanical Model) is significantly simplified, and the original section 2.3 is now in the appendix.
- Expanded conclusion to talk about current limitations of our approach and future work.
- Added a reference to Karl Sims "virtual creatures" work.
- Small typos

---

### Comment · Area_Chair_Bz5g · 2022-11-15
**Please engage before the author-reviewer discussion closes**

Dear authors and reviewers,

The first phase of the discussion period is about to close on November 18.

For authors, please make sure to submit your rebuttal by the deadline. Leave some time for the reviewers to read it and respond while you are still allowed to further engage with them. Interactions between authors and reviewers are very important for the quality of the review process, so please make sure to engage.

For reviewers, please try to acknowledge and respond to the authors' rebuttal while the discussion period is still open for them to further interact with you.

Thank you for your participation in the review process!

Best,
The AC

---

### Decision · Program_Chairs · 2023-01-20

**Decision:**

Accept: notable-top-25%

**Justification For Why Not Higher Score:**

None of the reviewers selected the highest scores of 9 or 10 for the paper. Additionally, the scope of the paper may not be broad or central enough to warrant an oral presentation. The reviewers describe the paper as a solid and enjoyable applied machine learning paper, but it may not be of interest to a large portion of the community.

**Justification For Why Not Lower Score:**

This is a very well executed application paper.

**Metareview: Summary, Strengths And Weaknesses:**

All four reviewers recommend accepting the paper (6-8-8-8). They find the paper to be highly novel and creative, and praise its clear and engaging writing. The reviewers note that the paper is accessible to readers who are unfamiliar with meta materials and associated physics. They highlight the creative neuromechanical model proposed in the paper, which works at the intersection of materials and neural networks. Experiments include both simulation and real-world (hardware) results, which is quite unusual for an ML paper. Overall, this contribution is a beautiful example of an applied machine learning paper.

**Note From Pc:**

if the above contains the word "oral" or "spotlight" please see: "oral" presentation means -> notable-top-5% and "spotlight" means -> notable-top-25%. As stated in our emails, we are disassociating presentation type from AC recommendations